# miR-671-5p Inhibition by MSI1 Promotes Glioblastoma Tumorigenesis via Radioresistance, Tumor Motility and Cancer Stem-like Cell Properties

**DOI:** 10.3390/biomedicines10010021

**Published:** 2021-12-23

**Authors:** Jang-Chun Lin, Chun-Yuan Kuo, Jo-Ting Tsai, Wei-Hsiu Liu

**Affiliations:** 1Department of Radiation Oncology, Shuang Ho Hospital, Taipei Medical University, Taipei 110301, Taiwan; 13451@s.tmu.edu.tw (J.-C.L.); 10637@s.tmu.edu.tw (C.-Y.K.); 10576@s.tmu.edu.tw (J.-T.T.); 2Department of Radiology, School of Medicine, College of Medicine, Taipei Medical University, Taipei 110301, Taiwan; 3School of Biomedical Engineering, College of Biomedical Engineering, Taipei Medical University, Taipei 110301, Taiwan; 4Department of Neurological Surgery, Tri-Service General Hospital and National Defense Medical Center, No. 325, Sec. 2, Cheng-Kung Road, Taipei 11490, Taiwan; 5Department of Surgery, School of Medicine, National Defense Medical Center, Taipei 11490, Taiwan

**Keywords:** glioblastoma multiforme, miR-671-5p, Musashi-1 (MSI1), radioresistance

## Abstract

MicroRNAs (miRNAs) could be potential biomarkers for glioblastoma multiforme (GBM) prognosis and response to therapeutic agents. We previously demonstrated that the cancer stem cell marker Musashi-1 (MSI1) is an RNA binding protein that promotes radioresistance by increasing downstream RNA stability. To identify that MSI1 interacts with miRNAs and attenuates their function, we also get candidate miRNAs from the mRNA seq by predicting with TargetScan software. miR-671-5p in GBM cells interacts with MSI1 by intersecting the precipitated miRNAs with the predicted miRNAs. Notably, overexpression of MSI1 reversed the inhibitory effect of miR-671-5p. The phenotype of miR-671-5p in GBM cells could affect radiosensitivity by modulating the posttranscriptional activity of STAT3. In addition, miR-671-5p could attenuate tumor migration and cancer stem cell (CSC) characteristics by repressing the posttranscriptional activity of TRAF2. MSI1 may regulate GBM radioresistance, CSCs and tumor motility through miR-671-5p inhibition to increasing STAT3 and TRAF2 presentation. In vivo, the GBM tumor size was inversely correlated with miR-671-5p expression, but tumorigenesis was promoted by STAT3 and TRAF2 activation in the miR-671-5p-positive GBM population. miR-671-5p could be activated as a novel therapeutic target for GBM and has potential application as a predictive biomarker of glioblastoma prognosis.

## 1. Introduction

The most lethal primary malignancy in adults is glioblastoma multiforme (GBM). Despite multimodal treatments consisting of surgical intervention and adjuvant concomitant chemoradiotherapy, the prognosis remains dismal [1]. The short survival time of GBM patients is caused by the inability to perform radical resection, tumors containing cancer stem cells (CSCs), and a high migration ability, which confer resistance to adjuvant therapy [2]. Musashi-1 (MSI1) is an RNA binding protein (RBP). It is not only a well-known neural stem cell marker but also a factor of radioresistance that acts by enhancing homologous recombination repair, cancer stem cell-like properties and tumor motility in GBM [3]. MSI1 was recently reported to directly target the 3′ UTR of its target mRNAs to suppress their posttranscriptional activity [4].

The non-coding RNA (ncRNA) is a group of RNAs that are unable to translate into proteins [5]. More and more studies have revealed that ncRNAs, including microRNAs (miRNAs), circular RNAs (circRNAs), and long non-coding RNAs (lncRNAs) perform a critical role in glioma carcinogenesis, especially in GBM. lncRNAs are a set of ncRNAs with more than 200 nucleotides and they can directly modulate tumorigenic molecules to induce higher SOX and lower p53 expression in GBM [6]. Actually, lncRNAs exist not only to promote but also to suppress GBM progression through cross-regulation of miRNAs. For example, CASC9 (a kind of lncRNA), miR-519d, and STAT3 make a positive circle to activate glioma carcinomgenesis [7]. On the other hand, circRNAs are a form of circle RNAs produced from pre-mRNA through back-splicing. *Barbagallo* et al. [8] determined that the expression level of circSMARCA5 more significantly downregulate in patient-derived GBM samples than controls. CircRNAs interact with miRNAs as miRNA sponges to hold miRNA suppressing mRNA. It seems no matter lncRNAs or circRNAs would maintain GBM pathophysiology via interacting with miRNAs [9]. MicroRNAs (miRNAs) are a group of short noncoding RNAs (~20–22 nucleotides) that regulate gene expression by binding to the 3′-untranslated region (3′ UTR) of target mRNAs [9]. They usually negatively regulate target genes by direct interaction with the 3ʹ UTR [10]. The aberrant expression of miRNAs is involved in multiple biological processes, such as cell proliferation, differentiation, and migration, and is associated with tumorigenesis and cancer progression [11,12]. miR-671-5p regulates tumor migration and stemness [13]. miR-671-5p has also been found to be abnormally expressed in many cancers, such as pediatric chordomas [14], glioma [15], breast cancer [16], and advanced rectal cancer [17]. However, the role of miR-671-5p in GBM is still elusive.

Previous literature has suggested a potentially critical role of MSI1 in GBM based on a small number of MSI1 targeted miRNAs [18]. In our study, we sought to identify the complete collection of MSI1 targeted miRNAs by using transcriptome-wide small RNA sequencing (RNA-seq). Our findings present evidence of miR-671-5p inhibition by MSI1. Furthermore, MSI1 inhibits miR-671-5p suppression of TNF receptor-associated factor 2 (TRAF2) to induce CSC characteristics and tumor invasion and silence signal transducer and activator of transcription 3 (STAT3) to enhance radioresistance in GBM. Therefore, our data indicate that the molecular mechanisms of glioblastoma tumorigenesis suggest potential therapeutic targets and prognostic markers.

## 2. Materials and Methods

**Chemicals and reagents.** Human glioblastoma U87MG cells were purchased from the Bioresource Collection and Research Center (Hsinchu City, Taiwan). 3-[4,5-Dimethylthiazol-2-yl]- 2,5-diphenyl tetrazolium bromide (MTT) (cat. no. M2128) was purchased from Sigma-Aldrich (St. Louis, MO, USA), Polyvinylidene difluoride (PVDF) membranes and an enhanced chemiluminescence solution (cat. no. WBKLS0500). TRIzol reagent (cat. no. 15596026) and Lipofectamine 3000 (cat. no. L3000015) were purchased from Invitrogen (Thermo Fisher Scientific). The plasmid for MSI1 overexpression was constructed in our previous study. SYBR1 Green PCR Master Mix (cat. no. 4309155), a MultiScribe^TM^ Reverse Transcriptase Kit (cat. no. N8080234), a TaqMan Advanced miRNA cDNA Synthesis Kit (cat. no. A28007), and TaqMan miR-671-5p (cat. no. 483035_mir), miR-23a-5p (cat. no. rno481350_mir), miR-29b-1-5p (cat. no. mmu482721_mir), miR-100-3p (cat. no. 478619_mir), miR-148a-5p (cat. no. 478718_mir), miR-190a-5p (cat. no. 478358_mir), miR-210-3p (cat. no. rno481343_mir), and miR-625-5p (cat. no. 479469_mir) miRNA assays were purchased from Applied Biosystems (Thermo Fisher Scientific). The dual-luciferase reporter assay system (cat. no. E1910) was purchased from Promega (Madison, WI, USA).

**Cell culture and transfection.** Hs683, SW1783, U251, and U87 GBM cell were purchased from ATCC. Normal brain samples were collected from our previous study [3]. The culture medium for maintaining cells was complement in an incubator with 5% CO_2_ at 37 °C with those mixture including serum fetal bovine (FBS 10%) getting from Biological Industries; antibiotics (penicillin + streptomycin), in addition, sodium pyruvate and amino acids. Human U87MG cells was kept in Eagle’s minimum essential medium. For conducting the transfection experiments, GBM cells should be planted into a 12-well plate with 10^5^ cells per well. When those cells in a well were 70% confluent growth, pcDNA6.2-miR-671-5p-EmGFP, p3XFLAG-MSI1, and 500 ng pmirGLO 3′ UTR reporter plasmids under the indicated doses made transfection with Lipofectamine 3000 (Invitrogen), according to the manufacturer’s instructions. After incubating for 24 h, The lysed cells were prepared for following study.

**MiRNA isolation, detection, and sequencing.** The small RNA fraction of MSI1-interacting RNAs was extracted with an miRNeasy Mini Kit (QIAGEN). For miRNA detection, to perform qRT-PCR with specific primer sets used Applied Biosystems (TaqMan miRNA assays). All protocols and reagents were obtained from TaqMan miRNA Biosystems. Using a 7900HT Fast RT-PCR System measures the following detection with an internal control, RNU6B. For qRT–PCR of specific miRNA, triplicate procedures were repeated on Applied Biosystems three times (Sequence-detecting system of ABI Prism 7900). Extracted miRNAs were used for library preparation from New England Biolabs product of a NEBNext^®^ Small RNA workflow. Illumina HiSeq 2000 analyzed extracted miRNAs. Sequences generated through the Illumina HiSeq 2000 Sequencing system to process for size exclusion of sequences < 15 nt with Cutadapt and adaptor removal. Filtered reads could be mapped to the hg38 assembly with Bowtie v1.0. [19], permitting 2 mismatches. It computed miRNA expression levels as the number of reads that miRNAs would be mapped in miRBase v18.

**Soft Agar Colony Formation Assay.** Planting cells were received ionizing radiation (IR) (3 Gy) and were then cultured for an additional two passages. A lower layer of agar (0.5%) was poured in complete DMEM (Gibco, NY, USA) and made to solidify, and an upper layer (0.35%) consisting 2.5 × 10^3^ U87MG cells suspended in a DMEM–agar mixture was then poured. After 14 days of incubation, counting colonies was performed. The relative numbers of formed colonies were determined from random microscopic fields on each plate. The colony-forming ability served as a malignant indicator of tumor cells. The average of three independent experimental data was presented.

**RNA-Seq Transcriptome analysis.** To use the DRAGEN pipeline obtains raw mRNA sequencing for RNA processing. The transcript per million (TPM) values were calculated with StringTie. In the preprocess of mRNA analysis, nonexpressed genes (TPM <  2) were excluded. Then, the other transcripts were used for the downstream analysis. Ballgown was used between the GBM and control samples to detect differentially expressed genes (DEGs) based on the expression levels got a threshold of an adjusted *p*-value less than 0.01 from StringTie. Different expression levels of miRNAs were identified by the DRAGEN RNA pipeline with a custom reference consisting of RNA-seq data from Ensemble. The identified miRNAs were obtained from the manufacturer’s instructions.

**Stable overexpression of miR-671-5p by a lentiviral vector.** To examine, in vivo and in vitro, the long-term expressed miR-671-5p, the vector with expression of pcDNA 6.2 GW and EmGFP-miR was inserted miR-671-5p. In brief, to synthesize and anneal oligos contain miR-671-5p stem-loop sequences, through T4 DNA ligase ligated annealed oligos. Those oligos should combine that vector with co-expression of pcDNA 6.2 GW and EmGFP-miR (Invitrogen). Lentiviral formation of transfection 293T cells was produced via the protocol of Lipofectamine 3000 after 5 × 10^6^ cells were planted in 10 cm plate. 48 h post transfection, the supernatants were collected and filtered. Cell sorting with fluorescence-activated analysis (FACS) was then used to confirm the viral titers post transduction at 48 h. Lentivirus infected into subconfluent cells in the existence of 8 μg/mL polybrene getting from Sigma-Aldrich. For making sure successfully infected GBM-CSCs, the co-expression cells of lentiviral infection, GFP, acted as a selection marker.

**The TRAF2 3′ UTR reporter plasmid formation.** The primers in PCR are listed in Appendix A. Specific primers for the TRAF2 3′ UTR in PCR was performed. The forward primer has a XhoI site and the reverse primer included a XbaI site. We used U87MG genomic DNA as a template and amplified the indicated region with primers containing restriction enzyme sites. We digested the DNA fragment with the corresponding endonucleases (Xho1 and Xba1) and cloned the digested fragment into the MCS region of pmirGLO (Promega). pmirGLO-TRAF2-3′ UTR was sequenced and named. For reporter assays, cells produce protocol of Lipofectamine 3000 transfection transiently with reporter plasmids, pmirGLO, and plasmids, miR-671-5p. The reporter experiment performs after transfection 24h later through a Dual-Luciferase Assay Kit from Promega, USA.

**Tumor cell transplantation and animals.** In vivo study, according to the Guidelines for Laboratory Animals all animals were bred and maintained in lab. of the Tri-Service General Hospital. To establish the subcutaneous mouse model, we needed to harvest, wash, and suspend GBM U87MG cells in PBS. U87 cells were transplanted into the male BALB/c nude eight week old mice. The average tumor volume calculates with the following equation: 1/2 × W^2^
× L = Volume (W, short diameter; L, long diameter). The measurement of the tumor size in the subcutaneous xenograft model performs every two days through using a caliper. 2 × 10^5^ U87MG cells/μL suspended from 1 × 10^6^ cells in 5 μL of PBS for orthotopic injection. Those eight-week-old NOD-SCID male mice (*n* = 6 per group; total, 36 mice) were injected with tumor cells according to the following coordinates using a stereotaxic apparatus: 2 mm below the dura; in the right of the bregma with 2 mm lateral and 3 mm posterior region. Tumor progression at five-day intervals was recorded using a 3T-MRI from Biospect system. The 3T-MRI signals accepted from radiofrequency transmission vis a miniquadrature coil.

**Statistical analysis.** The exact two-sided binomial test to analyze those clinical samples of miR-671-5p expression. The data are showed as the values of mean ± standard error (S.E.). Those results in MTT assay between mimic-transfected miR-671-5p groups and the control analyze via the statistical method of Permutation tests. Among the MSI1-, miR-671-5p-, miR-671-5p/MSI1-overexpressing and control groups analysis, one of the statistical methods, Student’s *t*-test analyzed to compare. Definition of statistically significance was *p*-values < 0.05.

The materials and methods used for the Transwell assay, TUNEL assay and TaqMan miRNA analysis are described in detail in the supplements.

## 3. Results

### 3.1. miR-671-5p Is Downregulated by MSI1 in Glioblastoma

To validate the essential role of miR-671-5p in normal brain and glioma cells, we first determined that miR-671-5p expression was significantly reduced in gliomas compared with human brain tissue by quantitative real-time PCR (qRT-PCR) (Figure 1a). Furthermore, in evaluating changes in the expression level of miR-671-5p from low-grade to high-grade glioma (Hs683, SW1783, U251, and U87MG) by qRT-PCR analysis, we found that the higher the grade of malignant glioma (U251 and U87MG), the lower the expression level of miR-671-5p (Figure 1b). On the other hand, MSI1 expression was higher in more malignant glioma cell lines (Figure 1c). Here, we investigated the connections between MSI1 and miRNAs in cancer stem cells in terms of features such as self-renewal, tumor-initiating ability, and radioresistance. In addition, our previous study demonstrated that MSI1 may promote gene expression by stabilizing mRNA [20].

Since MSI1 has RNA binding ability, we hypothesized that MSI1 might directly or indirectly bind to miRNAs and facilitate their function. To prove this concept, we used small RNA-seq to detect differences in miRNA expression after MSI1 overexpression. Based on the bioinformatics analysis results, we focused on several abundant and significantly altered miRNAs (mean intensity in the control group > six times the mean intensity in the test group, log(fold change) > 2.5), including miR-29a-5p, miR-33b-3p, miR-185-3p, miR-190-5p, miR-194-3p, miR-625-5p, miR-671-5p, and miR-4521. Further confirming the expression trend, the qRT-PCR results revealed that MSI1 overexpression significantly decreased the expression of miR-671-5p and miR-4521 in U87MG cells. Finally, we obtain miR-671-5p from the qPCR results because miR-671-5P has the most significant change. (Figure 1d). Among the downregulated miRNAs, miR-671-5p was the most downregulated (0.14 ± 0.16, *p* = 0.00211). We will expression profile in the Appendix A. In addition, a recent study has reported that miR-671-5p has a crucial role in the brain. For example, Benjamin Kleaveland et al. demonstrate miR-671-5p can for a regulatory network in the brain [21], which has increasingly appreciated gene-regulatory roles. Dysregulated miR-671-5p is also involved in glioblastoma multiforme [15]. Of note, miR-671-5p inhibits tumor cell proliferation and promotes cell apoptosis [22]. This evidence convinces us to choose mir-671-5p instead of other miRNAs. Next, to investigate whether MSI1 interacts with miR-671-5p in vitro, we overexpressed MSI1 (MSI1-OE) in U87MG cells, and the qPCR results showed that MSI1 overexpression reduced miR-671-5p expression (Figure 1e). At the same time, the level of miR-671-5p increased when MSI1 was knocked down (MSI1-KD) (Figure 1f). RNA-IP analysis of MSI1-interacted miR-671-5p compared to that in IgG group. The results determine that MSI1 could directly interact with miR-671-5p (Figure 1g). Taken together, these data indicate that MSI1 inhibits miR-671-5p expression.

### 3.2. miR-671-5p Promotes Radiosensitivity by Suppressing DNA Repair, Cancer Stem-like Properties and Tumor Migration

Previous studies have reported that miR-671-5p may inhibit epithelial-to-mesenchymal transition and promote apoptosis [23]. However, whether miR-671-5p plays an anti-oncogenic role in GBM remains unclear. We inserted miR-671-5p into the pcDNA3.2-miRNA-eGFP plasmid, transfected the plasmid into U87MG cells, and finally selected transfected cells with puromycin. Transfection of a plasmid encoding enhanced green fluorescent protein (eGFP) in vitro enabled us to detect the transfected cells in vitro (Figure 2a, left). The qRT-PCR results showed that miR-671-5p was highly expressed in transfected cells compared to control cells (Figure 2a, right). To identify the biological role of miR-671-5p in glioblastoma, we used RNA-seq to determine the transcriptome after miR-671-5p overexpression and used gene set enrichment analysis (GSEA) to reveal their candidate pathways. GSEA showed that DNA repair (NES, 0.94) and epithelial-mesenchymal transition (EMT) (NES, 0.920) were enriched in the miR-671-5p overexpression group (Figure 2b). Furthermore, we used a heatmap to illustrate the significant genes in each pathway under miR-671-5p overexpression conditions (Figure 2c). The result implies that miR-671-5p has a negative correlation with DNA repair and EMT activation.

The TUNEL assay detects early DNA damage to determine apoptotic cell death. The percentages of TUNEL-positive cells in the control and miR-671-5p overexpression groups without radiation exposure were 0.73 ± 0.08% and 0.92 ± 0.25%, respectively, whereas those in the same groups with exposure to 3 Gy radiation were 1.82 ± 0.43% and 5.33 ± 4.1% (Figure 2d). The TUNEL assay revealed that miR-671-5p overexpression significantly increased apoptosis (*p* = 0.00154). Consistent with this finding, we confirmed that miR-671-5p enhanced the radiation response in U87MG cell lines. In the sphere formation assay, we found a significant reduction in the glioblastoma spheroid formation ability upon miR-671-5p overexpression (Figure 2e, left) and performed further quantification of sphere formation in both the U87MG and U251 cell lines (Figure 2e, right). In the migration assay, fewer U87MG cells migrated in the miR-671-5p overexpression group than in the control group (Figure 2f, left), and quantification by ImageJ software showed significantly different numbers of migrated cells in the miR-671-5p-overexpressing and control cells (Figure 2f, right). Concerning the relationships between miR-671-5p and EMT markers, including E-cadherin (E-CAD), N-cadherin (N-CAD), SNAIL, SMA, and TWIST mRNA expression, qRT-PCR results showed that miR-671-5p significantly suppressed the expression of N-CAD, SNAIL, and TWIST but increased the E-CAD level (Figure 2g). To further validate the protein level are also affected by miR-671-5p, we harvested the cell lysate of U251 and U87 cells for western blot. The western blot result revealed that the mesenchymal markers, including N-CAD, SNAIL, and TWIST, are increased (Appendix A). Higher E-CAD expression induces stationary EMT without inducing tumor motility. Therefore, according to a series of phenotypic data, miR-671-5p in glioblastoma cells enhances radiosensitivity by increasing apoptosis and reducing DNA repair, tumor migration and stem cell-like abilities.

### 3.3. Activation of the MSI1/miR-671-5p/STAT3 Axis Regulates Radiosensitivity

STAT3 promotes glioblastoma stem-like properties and is a critical mediator of resistance to irradiation [2] Based on enrichment analysis (Figure 2c), we reasoned that overexpression of miR-671-5p could sensitize cells to radiotherapy by reducing the cell recovery ability of STAT3. In U87MG cells, miR-671-5p reduced STAT3 expression, as shown by qRT-PCR (Figure 3a). Meanwhile we also confirmed the protein level of STAT3 in the presence of miR-671-5p. The western blot result showed that the STAT3 are decreased in the presence of miR-671-5p (Appendix A). To further investigate the long-term effect of miR-671-5p on the radiation response, we used a soft agar colony formation assay to monitor whether miR-671-5p affects cell viability after irradiation. The miR-671-5p-, STAT3-, and miR-671-5p/STAT3-overexpressing cells were irradiated at 3 Gy and cultured for an additional two passages. Significantly decreased colony formation after irradiation was observed in miR-671-5p-overexpressing U87MG cells (Figure 3b, upper right), but the suppressive effect of irradiation on colony formation was abolished in miR-671-5p/STAT3 double-overexpressing cells (Figure 3b, right lower). The quantitative colony formation results indicated that miR-671-5p significantly reduced the colony number (33.0 ± 3.0 vs. 10.0 ± 4.0, *p* = 0.00211) (Figure 3c). Furthermore, an apoptosis assay with Annexin V staining showed that miR-671-5p/STAT3 overexpression induced apoptosis escape in U87MG cells after irradiation (Figure 3d).

To support the hypothesis of miR-671-5p-mediated sensitization, we used an alkaline comet assay to investigate the effect of miR-671-5p and miR-671-5p/STAT3 expression on radiation damage. In the alkaline comet assay, the shifted portion of the fluorescence image indicates the severity of DNA damage (Figure 3e, left). The comet assay showed that miR-671-5p overexpression significantly increased DNA damage (53.3 ± 4.4 vs. 5.2 ±1.4, *p* = 0.00131). Notably, STAT3 overexpression reduced DNA damage in miR-671-5p-overexpressing U87MG cells (Figure 3e, right). Consistent with this finding, we confirmed that miR-671-5p sensitized the cells to radiation. To investigate whether miR-671-5p can inhibit the posttranscriptional regulation of STAT3, we incorporated the STAT3 3′ UTR into the pmirGLO reporter plasmid (Figure 3f, left), which was designed to quantitatively evaluate miRNA activity. The STAT3 3′ UTR has an miR-671-5p binding site, as indicated by the dashed line in Figure 3f. The luciferase assay revealed that transfection with a larger concentration of miR-671-5p decreased luciferase activity, whereas transfection with the control plasmid showed a significant inhibitory effect on STAT3 3′ UTR activity (0.367 ± 0.08 vs. 0.788 ± 0.11, *p* = 0.0021), with a dose-dependent effect of MSI1 transfection (Figure 3f, right). Therefore, STAT3 might maintain tumor growth after irradiation through MSI1-mediated attenuation of miR-671-5p.

### 3.4. The MSI1/miR-671-5p/TRAF2 Axis Mediates EMT and CSC Abilities

Based on our previous finding that glioblastoma stem-like cells can possess higher tumor invasion and migration abilities [3], we became interested in whether miR-671-5p affects cell motility through attenuation of EMT-related proteins. Thus, based on the enrichment analysis results (Figure 2c), the expression levels of EMT-associated genes in miR-671-5p- and miR-671-5p+MSI1-overexpressing cells were compared. Significantly lower expression of TRAF2 mRNA was found in cells overexpressing miR-671-5p alone than in cells overexpressing miR-671-5p+MSI1 (Figure 4a). To further investigate whether miR-671-5p targets the TRAF2 3′ UTR, we incorporated the TRAF2 3′ UTR into pmirGLO and named the resulting plasmid pmirGLO-TRAF2–3′ UTR (Figure 4b, upper). The luciferase assay results (Figure 4b, lower) showed a dose-dependent effect of MSI1 transfection in rescuing TRAF2 3′ UTR luciferase activity in the presence of miR-671-5p (1.00 ± 0.123 vs. 0.455 ± 0.09, *p* = 0.00345). Transfection of the miR-671-5p-emGFP plasmid encoding GFP allowed us to monitor miR-671-5p expression in vitro. To observe whether MSI1 or miR-671-5p affects TRAF2 expression, we performed immunofluorescence staining for TRAF2 in U87MG cells. Consistent with each other, the immunofluorescence and qRT-PCR results showed that miR-671-5p overexpression significantly reduced TRAF2 expression (1.00 ± 0.09 vs. 0.47 ± 0.06, *p* = 0.00531) (Figure 4c). Then, in evaluating the regulation of CSC properties by the MSI1/miR-671-5p/TRAF2 axis, we found that CSC markers in GBM cells with overexpression of miR-671-5p alone were significantly reduced compared with those in cells with TRAF2 overexpression, and this effect was rescued by co-overexpression of MSI1 and miR-671-5p, consistent with the mRNA level quantification results (Figure 4d,e).

The migration differences among MSI1, miR-671-5p, miR-671-5p/MSI1 and control cells with TRAF2 overexpression were assessed by using a Transwell migration assay. The results showed that miR-671-5p overexpression reduced the number of cells invading the basement membrane (Figure 4f). Moreover, we found that TRAF2 overexpression significantly rescued the migration ability in the presence of miR-671-5p (54.0 ± 7.0 vs. 21.0 ± 3.0, *p* = 0.00401). Although TRAF2 has less evidence in the GBM, there are several reported supporting that TRAF2 associated with migration of gastric cancer cells by regulating TRAF2 [24]. Overall, we concluded that MSI1 maintains EMT properties by inhibiting the miR-671-5p/TRAF2 axis.

### 3.5. miR-671-5p Is Inversely Related to Glioblastoma Tumorigenesis In Vivo

To investigate the relevance of our findings in an animal model, we transplanted MSI1-, miR-671-5p-, and MSI1/miR-671-5p-overexpressing cells into nude mice. However, MSI1 overexpression did not affect the transcription of miR-671-5p. To mimic the physiological environment of brain tumors, we transplanted U87MG cells into the brain. The MRI results showed that the tumor volumes in the miR-671-5p, miR-671-5p/MSI1, and control groups were 11.4 ± 2.1, 15.5 ± 3.2, and 20.3 ± 4.3 mm^3^, respectively. We observed a low growth rate of the miR-671-5p-overexpressing tumors, whereas the miR-671-5p/MSI1 and control tumors showed a higher growth rate (11.4 ± 2.1 vs. 20.3 ± 4.3 mm^3^, *p* = 0.00546) (Figure 5a). We tested the relationship between the growth rate and miR-671-5p- and TRAF2-overexpression in tumors by using a xenograft model. We also examined TRAF2 and miR-671-5p expression in an available tissue sample. TRAF2 and miR-671-5p expression was analyzed with qRT-PCR, which confirmed that the expression levels were consistent with those in the cells before transplantation. The qRT-PCR results showed that miR-671-5p was significantly upregulated in miR-671-5p-overexpressing cells (1.03 ± 4.77 vs. 0.17 ± 0.72, *p* = 0.00031). This finding indicates the stabilization of miR-671-5p expression in the in vivo models (Figure 5b). In the xenograft model, tumor size was measured three times weekly throughout the study. We observed significant differences in overall tumor growth rates between each group over time. Mice in the miR-671-5p-overexpressing group had a significantly slower tumor growth rate than those in the control group (*p* = 0.00943). Furthermore, the miR-671-5p/TRAF2 group had a significantly higher tumor growth rate than the miR-671-5p group (*p* = 0.00943), suggesting that TRAF2 might overcome the biological effects of miR-671-5p (Figure 5c). We also observed the survival rates in the STAT3, miR-671-5p, miR-671-5p/STAT and control groups of NOD/SCID mice after irradiation. miR-671-5p overexpression increased the survival rate of NOD/SCID mice. Notably, STAT3 overexpression decreased the survival rate of mice with miR-671-5p-overexpressing tumors, supporting that STAT3 has an inhibitory effect on miR-671-5p expression (Figure 5d). Overall, these findings indicate that MSI1 reversed the effect of miR-671-5p through activation of STAT3 and TRAF2 in the in vivo animal model.

### 3.6. miR-671-5p Is Inversely Correlated with STAT3 Activation and TRAF2 Presentation in Clinical GBM

Since we previously determined that both TRAF2 and STAT3 are suppressed by miR-671-5p to affect tumor migration, CSC ability and radioresistance in GBM, we attempted to identify the clinical connection between TRAF2 and STAT3 expression and disease progression. To seek clinical evidence from population-based statistics, we collected data for GBM patients from the TCGA database and identified MSI1-correlated genes. First, Pearson correlation analysis showed that the levels of both TRAF2 (*p* = 2.7 × 10^−11^; R: 0.49) and STAT3 (*p* = 4.0 × 10^−5^; R = 0.32) were positively correlated with that of MSI1 in GBM patients (Figure 6a). To further examine the impacts of these genes on clinical outcomes, we also assessed the survival rates of patients with different levels of TRAF2 and STAT3 expression. These results showed that the hazard ratios (HRs) of TRAF2 and STAT3 were 1.4 (*p* = 0.062) and 1.2 (*p* = 0.39), respectively (Figure 6b), indicating that TRAF2-positive patients showed poorer survival than TRAF2-negative patients. We also analyzed clinical GBM tissues to compare transcriptional differences. The qRT-PCR results showed that the expression of TRAF2, STAT3 and MSI1 was significantly upregulated in GBM patients (Figure 6c). In contrast, miR-671-5p expression was downregulated. Taken together, these data indicate that miR-671-5p expression in clinical samples is compatible with that in the in vitro cell model, implying that miR-671-5p suppresses the synthesis of proteins such as STAT3 and TRAF2. Therefore, the MSI1/miR-671-5p/STAT3 axis enhances radioresistance, and the MSI1/miR-671-5p/TRAF2 axis activates glioblastoma stem cell-like properties and EMT-like abilities (Figure 6d).

## 4. Discussion

Recently, several studies have shown that miRNAs are critical molecular players closely related to the biological features of cancer stem cells. It is estimated that miRNAs can regulate up to 60% of human genes, including genes associated with maintaining the stem-like phenotype, differentiation, and chemo- and radioresistance [25,26,27]. Glioblastoma stem-like cells are a vital contributor to poor responses to adjuvant therapy due to their higher expression of DNA repair enzymes, antiapoptotic factors, and multidrug transporters [28]. Compelling evidence has revealed that the presence of CSCs, as determined by functional assays and the expression of glioblastoma stem cell markers, is associated with the prognosis of GBM patients [29,30]. MSI1 also cooperates with LIN28 to inhibit the posttranscriptional biogenesis of miRNAs in embryonic stem cells [31]. In our previous study [3], MSI1 acted as a glioblastoma stem-like cell marker and promoted tumor migration and radioresistance. This study further determined the downstream miRNA of MSI1 that regulates CSC properties, EMT-like abilities and radioresistance by establishing a collection of data including miRNA expression and downstream gene expression and clinical data derived from our GBM studies encompassing RNA-seq and small RNA-seq expression datasets. The findings indicate that miR-671-5p can be downregulated by MSI1 to attenuate the effect of miR-671-5p on the radiation response and tumorigenesis.

Two major signals from our study, STAT3 and TRAF2, are involved in miR-671-5p-regulated tumor migration. STAT3 are associated with glioblastoma invasion through Stat3/fibronectin pathway [32]. The cooperation of NF-κB and Stat3 recruitment regulates the radiation-induced invasion and migration in glioma [33]. Although TRAF2 has less evidence in the GBM, several reports support that TRAF2 is associated with migration of gastric cancer cells by regulating TRAF2 [24].

The current aim of human GBM therapy is to prevent cancer recurrence and delay progression. Drug resistance and radioresistance are essential factors for cancer recurrence. Prompt GBM detection and intervention improve the patient survival rate. A large amount of evidence has indicated that miRNAs play essential roles in the development of several types of human cancer, such as GBM [34]. Previous research has revealed that miR-671-5p plays a critical role in human cancer progression. A study by Barbagallo et al. reported that miR-671-5p is involved in GBM development [15]. miR-671-5p has been reported to promote cancer development and metastasis. miR-671-5p is upregulated in metastatic prostate cancer and accelerates the proliferation, migration, and invasion of colon cancer cells. High expression of miR-671-5p is often associated with poor prognosis in prostate and colon cancer patients. miRNA-671-5p can increase cell proliferation, migration, and invasion in vitro and in vivo [35]. Interestingly, a previous study revealed that miR-671-5p is a potential antioncogene in breast cancer. Decreased expression of hsa-miR-671-5p is correlated with unfavorable survival outcomes, invasion, microsatellite formation, and reduced differentiation. In vitro experiments of miR-671-5p overexpression in breast cancer revealed reduced cell proliferation, invasion, and radioresistance [36]. From our results, we first identified the roles and underlying mechanisms of MSI1 and miR-671-5p in GBM, which regulate radioresistance by STAT3 inhibition and cancer migration and CSC function by TRAF2 suppression.

This study first discovered MSI1-interacting miRNAs and used a strategy to filter potential mediators in GBM cells. We successfully demonstrated that MSI1 directly reverses the inhibitory effects of miR-671-5p via mechanisms including TRAF2 and STAT3 dysregulation. A solid causal relationship between miR-671-5p and MSI1 expression was demonstrated in a panel of GBM cancer cell lines. We also found strong correlations among STAT3, TRAF2, and MSI1 in clinical samples. Collectively, our data strongly suggest that miR-671-5p plays a vital role in maintaining downstream STAT3 and TRAF2 expression. Our in vitro studies demonstrated that MSI1 overexpression reversed miR-671-5p-mediated tumor growth, DNA repair, and tumorsphere growth. Notably, the miR-671-5p level was significantly reduced in GBM tumor tissue compared to paired adjacent normal tissue. Restoration of MSI1 expression in xenograft tumor models coincidentally increased tumor growth in vivo. Our work revealed a novel mechanism of the MSI1/miR-671-5p/TRAF2 and MSI1/miR-671-5p/STAT3 axis, providing therapeutic targets for GBM.

## Figures and Tables

**Figure 1 biomedicines-10-00021-f001:**
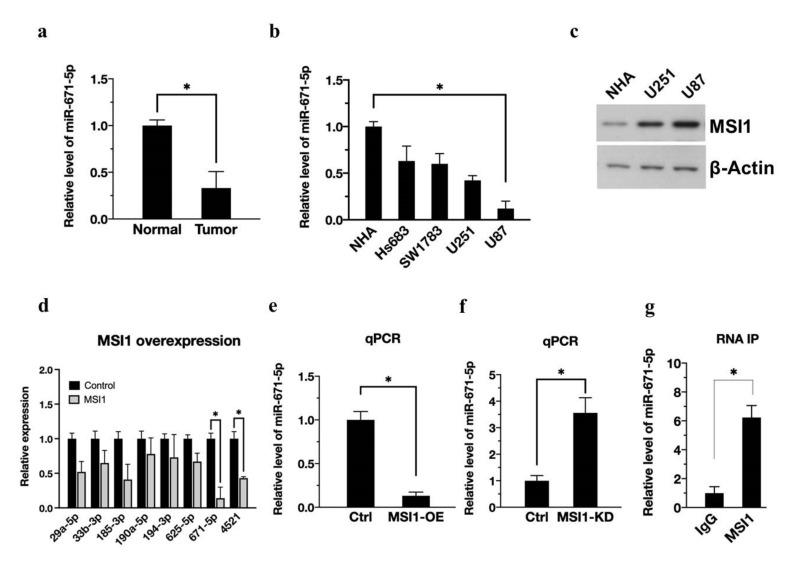
MSI1 activation downregulates miR-671-5p expression. (**a**). qRT-PCR analysis of expression of miR-671-5p in clinical tumor cells compared to the adjacent normal group. The miR-671-5p level is presented as a mean from three independent experiments with SD error bars. An asterisk indicates a statistically significant difference (Student’s *t*-test, *p* < 0.01). (**b**) qRT-PCR analysis of expression of miR-671-5p in Hs683, SW1783, U251, and U87MG cells compared to control NHA group. (**c**) Western blotting analysis of expression of MSI1 in NHA, U251, and U87MG cells. (**d**) qRT-PCR analysis of expression of miRNAs in MSI1-overexpressed U87MG cells compared to control group. (**e**) qRT-PCR analysis of expression of miR-671-5p in MSI1-overexpressed or (**f**) MSI1-knockdown U87MG cells compared to control cells. (**g**) RNA-IP analysis of MSI1-interacted miR-671-5p compared to that in the IgG group. The miR-671-5p level is quantified relative to the IgG group and presented as a mean from three independent experiments with SD error bars. An asterisk indicates a statistically significant difference (Student’s *t*-test, * *p* < 0.01).

**Figure 2 biomedicines-10-00021-f002:**
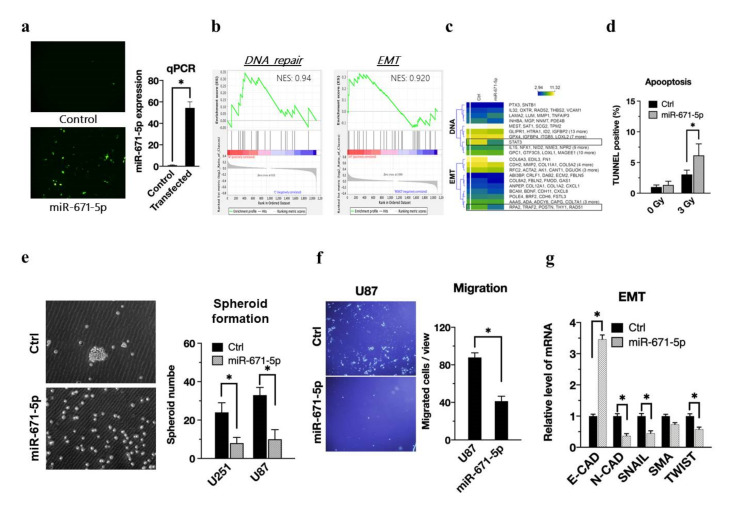
miR-671-5p associates with reducing radioresistance, cancer stem-like cell property and epithelial-mesenchymal transition. (**a**, left) Representative image of miR-671-5p-transfected cells with emGFP labeling. The emGFP-positive cells were compared to the control group and presented as a mean from three independent experiments with SD error bars. (**a**, right) qRT-PCR analysis of miR-671-5p expression in GBM U87MG cells compared to control group. (**b**) Gene Set Enrichment Analysis of all affected genes by miR-671-5p overexpression (fold change >2 and <0.5). NES, normalized Enrichment Score. (**c**) The heatmap of individual genes including DNA repair and EMT associated genes from the GSEA analysis. Yellow: the upregulated genes; blue: the downregulated genes. (**d**)TUNEL assay profiling apoptosis status of control(ctrl), and miR-671-5p- overexpressed cells after 0 or 3 Gy irradiation. TUNEL positive cells were compared to the control group and presented as a mean from three independent experiments with SD error bars. (**e**) The spheroid formation assay of MSI1-overexpressed GBM U87MG cells. The spheroid number were compared to the control group and presented as a mean from three independent experiments with SD error bars. (**f**) Migration assay profiling migration ability of miR-671-5p-overexpressed cells. The migrated cell number was quantified into a box plot. (**g**) qRT-PCR analysis of expression of EMT associated genes in U87MG cells. GAPDH was used as a loading control. The mRNA level is compared to the control group, quantified relative to the vehicle group, and presented as a mean from three independent experiments with SD error bars. Asterisk indicates a statistically significant difference (Student’s *t*-test, * *p* < 0.01).

**Figure 3 biomedicines-10-00021-f003:**
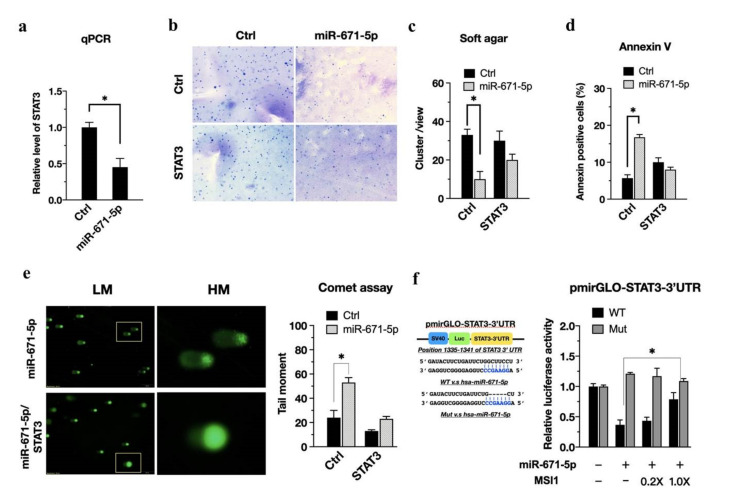
miR-671-5p suppresses STAT3 to activate radiosensitivity, but MSI1 rescues U87MG cells via inhibition of this mir-671-5p/STAT3 axis. (**a**) qRT-PCR analysis of STAT3 expression in miR-671-5p-overexpressed U87MG cells. GAPDH was used as a loading control. The mRNA level is compared to the control group, quantified relative to the vehicle group, and presented as a mean from three independent experiments with SD error bars. An asterisk indicates a statistically significant difference (Student’s *t*-test, *p* < 0.01). (**b**) Soft agar colony-formation assay of U87MG cells transfected with miR-671-5p, STAT3, miR-671-5p/STAT3, or control plasmid, and (**c**) quantification of the colony number in each group. (**d**) Flow cytometry annexin V assay and (**e**) comet assay of control, miR-671-5p-, STAT3, miR-671-5p/STAT3 overexpressed cells after 3 Gy irradiation. (**f**, left) A scheme presentation of pmirGLO-STAT3-3′UTR map. The 3′UTR of STAT3 has the binding sites for miR-671-5p. (**f**, right), luciferase reporter assay for the direct interaction of miR-671-5p with predicted target sites in the 3′UTRs of STAT3. The MSI1 overexpression attenuated the inhibition of luciferase activities by miR-671-5p. The U87MG cells were transfected with different doses of MSI1 plasmid (empty, 0.2X, and 1X). The luciferase activity is normalized to Renilla and presented as the mean from three independent experiments with SD error bars. An asterisk indicates a statistically significant difference (Student’s *t*-test, * *p* < 0.01).

**Figure 4 biomedicines-10-00021-f004:**
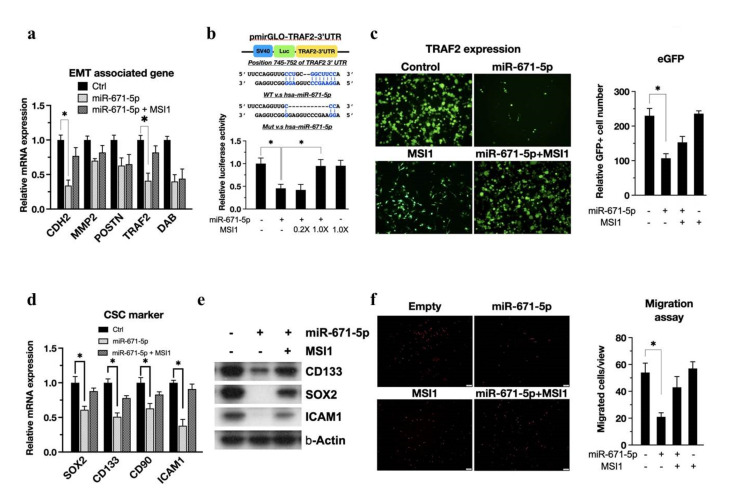
miR-671-5p inhibits the TRAF2 post−transcription. (**a**) qRT−PCR analysis of EMT associated genes in miR-671-5p−and/or MSI1− overexpressed GBM U87MG cells compared to control group. The mRNA level is quantified relative to the vehicle group and presented as the mean from three independent experiments with SD error bars. Asterisk indicates a statistically significant difference (Student’s *t*-test, *p* < 0.01). (**b**, upper) A scheme presentation of pmirGLO-TRAF2-3′UTR map. The 3′UTR of TRAF2 is matching the seed region of miR-671-5p; (b, lower) luciferase assay of pmirGLO-STAT3-3′UTR in miR-671-5p-, 0.2X MSI1-/miR-671-5p-, 1.0X MSI1-/miR-671-5p-, and 1.0X MSI1- transfected U87MG cells compared to vehicle control. The luciferase activity is normalized to Renilla and presented as the mean from three independent experiments with SD error bars. (**c**) The immunofluorescence of TRAF2 in miR-671-5p-, MSI1-, MSI1-/miR-671-5p-overexpressed, and control U87MG cells. (**d**) qRT-PCR analysis of CSC marker genes in miR-671-5p- and/or MSI1- overexpressed U87MG cells compared to control group. (**e**) Western blotting of CSC marker in miR-671-5p and/or MSI1 transfected U87MG cells. (**f**, left) Migration assay profiling migration ability of miR-671-5p-, MSI1-, MSI1-/miR-671-5p-overexpressed, and control U87MG cells. (**f**, right) The migrated cell number was quantified into a box plot.

**Figure 5 biomedicines-10-00021-f005:**
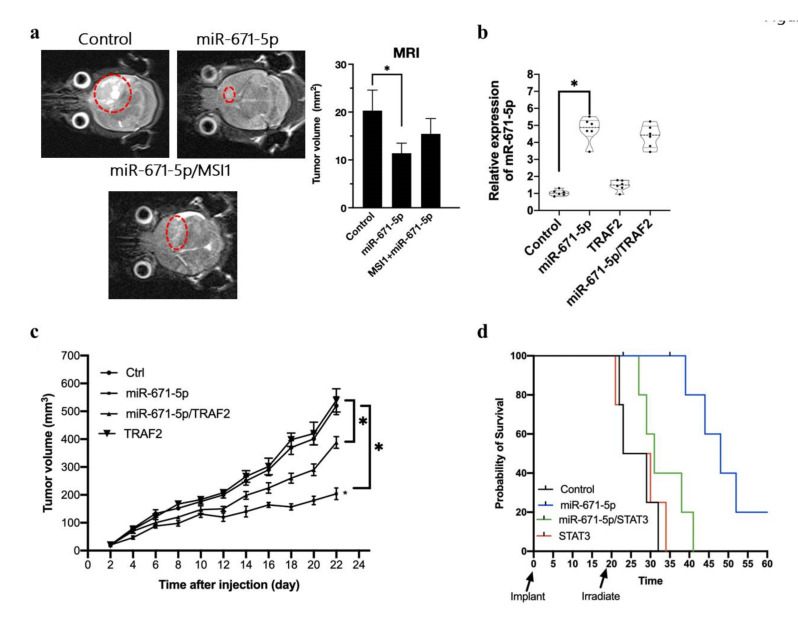
MSI1 attenuates the miR-671-5p mediated inhibition in tumour size. (**a**) 3T-MRI photography showing the transplanted tumor in Nude/SCID mice. The transplanted tumor is labelled with the dotted circle. Besides the subcutaneous transplantation, we also transplanted the miR-671-5p, miR-671-5p/MSI1 overexpressed and control U87MG cells into the brain and assayed the survival rate of mice. (**b**) qRT-PCR analysis of miR-671-5p expression of transplanted miR-671-5p, TRAF2, miR-671-5p/TRAF2, and control tissue in mice. The mRNA level is quantified relative to the control group and presented as the mean from three independent experiments with SD error bars. An asterisk indicates a statistically significant difference (Student’s *t*-test, *p* < 0.01). (**c**) The tumor growth rate was measured every other day. Variations in tumor volume (mm3) overtime in four treatment groups: miR-671-5p, TRAF2, miR-671-5p/TRAF2, and control cells. All groups were irradiated (3 Gy) before transplantation and culture for two passages and then transplanted into mice. Twenty-two days after transplantation, we harvested all the samples for further analysis. Error bars represent standard deviation (SD). The symbol denotes significance (*p* < 0.001) relative to control. (**d**) The survival rate analysis of miR-671-5p, STAT3, miR-671-5p/STAT3 overexpressed and control cells after irradiation. We irradiated the mice on day 20 and observed the survival rate within 60 days.

**Figure 6 biomedicines-10-00021-f006:**
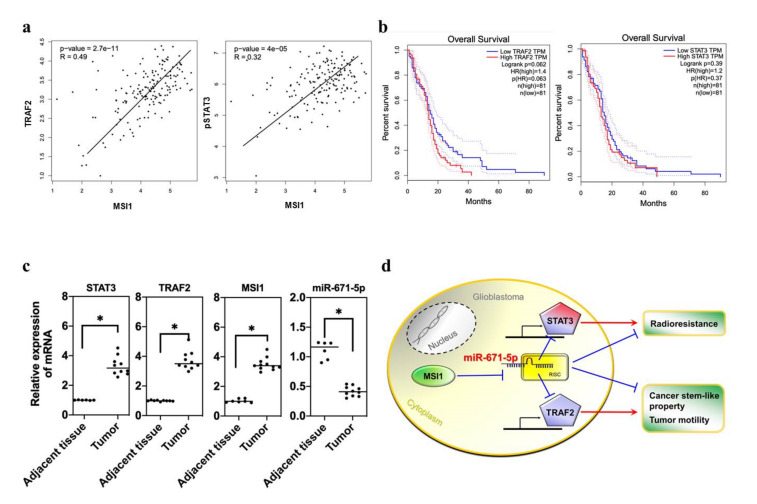
The clinical importance of MSI1-/miR-671-5p-associated genes in GBM patients. (**a**) Correlation Analysis of MSI1/STAT3 and MSI1/STAT3 in GBM patients. (**b**) The overall survival rate of TRAF2 or STAT3 regulation network of a candidate gene in GBM patients. (**c**) qRT-PCR analysis of expression of STAT3 or TRAF2 mRNA in GBM tissues compared to adjacent normal tissues. The mRNA level is quantified relative to the vehicle group and presented as the mean from three independent experiments with SD error bars. An asterisk indicates a statistically significant difference (Student’s *t*-test, *p* < 0.01). (**d**) The illustration of the mechanism that miR-671-5p suppresses the radioresistance by inhibiting STAT3 and inhibits cancer stem cell properties and tumor motility by inhibiting TRAF2 in GBM cells.

## Data Availability

The datasets used and/or analyzed during the current study are available from the corresponding author on request. This will in most cases also require an ethical permit.

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
