# Peer review of "miR-671-5p Inhibition by MSI1 Promotes Glioblastoma Tumorigenesis via Radioresistance, Tumor Motility and Cancer Stem-like Cell Properties"

_biomedicines, 2021, doi:10.3390/biomedicines10010021_

Round 1

Reviewer 1 Report

  1. Figure 1 showed the MSI1 downregulate various miRNAs. Please explain why only miR-671-5p was selected to investigate in the study. In addition, please introduce the relationships between tumor cells and these miRNAs (Fig. 1D).
  2. Messenger RNA levels are not equal to protein levels. In figure 2g, only EMT-related mRNAs were determined. Authors have to show the EMT-related protein levels. In addition, only STAT3 mRNA was be observed in figure 3A, authors must provide the STAT3 protein levels.
  3. Please mention which signals were involved in miR-671-5p-regulated tumor migration.
  4. Please introduce which miRNAs can be regulate by MSI1. In addition, whether these miRNAs can interact with miR-671-5p?

Reviewer 2 Report

In this manuscript, the authors have investigated the effects of MSI1, an RNA binding protein, on miR-671-5p, a micro RNA that plays a tole in tumorigenesis, cell proliferation and cancer progression. The authors sought to investigate and identified the complete pathway axis through which MSI1 and miR-671-5p affects the cancer progression. The authors concluded that the miR-671-5p was inhibited by MSI1. MSI1 increase was also observed in GBM, thus MSI1 aid in cancer progression by miR-671-5p. MSI1 also induce TNF receptor associated factor 2 (TRAF2) via inhibition of miR-671-5p to induce CSC characteristics and tumor invasion. The work published in this manuscript shed lights on MSI1/miR-671-5p/TRAF2 and MSI1/miR671-5p/STAT3 axis in GBM, providing a therapeutic targets for GBM.

The manuscript is well written and provides enough background. The data presented in it also supports the claims made by the authors. Overall, the manuscript can be accepted in its current form.

One very minor change that needs to be made is that in the last paragraph (Before material and methods), In line-4, it is mentioned as "miR-671-50". This should have been "miR-671-5p".

Reviewer 3 Report

Lin and colleagues proposed an interesting research article aimed at elucidating the molecular interaction between MSI1 and miR-671-5p responsible for radioresistance in GBM. For this purpose, the authors have investigated the expression levels of these two factors together with other associated genes in order to establish the role of miRNA and MSI1 in GBM resistance. Overall, the manuscript is interesting and performed in a rigorous manner. However, some experiments were not correctly described and the entire study is based on experimental observations obtained in a single cell line. Below are reported some major comments that will improve the manuscript:
1) In the Introduction section, please briefly mention that different studies have investigated the role of ncRNA (including microRNA, circRNA and lncRNA) as biomarkers for GBM and then describe the role of miR-671-5p. For this purpose, please see:
- PMID: 31322245
- PMID: 33126510
- PMID: 34198978
- PMID: 30583549
2) The results shown in the first paragraph of chapter 3.1 are unclear. In the M&M section, the authors described only U87MG cells while the results showed in Figure 1 are related to other cells (Hs683, SW1783, U251, and U87) and GBM and normal brain samples. Have the authors recruited patients? Please better clarify these experiments;
3) The entire manuscript is based on the experiments performed on a single GBM cell line. The authors have to confirm their results at least in an additional GBM cell line; 
4) According to the authors’ hypothesis, MSI1 could be able to bind microRNAs. Usually, miRNAs are able to bind and inhibit mRNA expression. Thus, the rationale of the study appears weak. Please argue this point;
5) From the Results section, it is not clear why the authors chose miR-671-5p instead of other miRNAs. Please better describe this part;
6) Throughout the manuscript please be consistent with the use of “U87MG”.

Round 2

Reviewer 1 Report

The manuscript has revised. 

Reviewer 3 Report

I have really appreciated the authors' answers. In particular, the authors well-addressed all my previous comments adding novel results on different cell lines and clarifying some doubtful aspects. The revised version of the manuscript is now more completed and detailed. The manuscript can be accepted for publication after the editorial check.